# Mechanisms Limiting Renal Tissue Protection and Repair in Glomerulonephritis

**DOI:** 10.3390/ijms24098318

**Published:** 2023-05-05

**Authors:** Andrea Angeletti, Maurizio Bruschi, Xuliana Kajana, Sonia Spinelli, Enrico Verrina, Francesca Lugani, Gialuca Caridi, Corrado Murtas, Giovanni Candiano, Marco Prunotto, Gian Marco Ghiggeri

**Affiliations:** 1Nephrology, Dialysis and Transplantation Unit, IRCCS, Istituto GianninaGaslini, 16147 Genova, Italy; andreaangeletti@gaslini.org (A.A.); mauriziobruschi@gaslini.org (M.B.); enricoverrina@gaslini.org (E.V.); francescalugani@gaslini.org (F.L.); gianlucacaridi@gaslini.org (G.C.);; 2Department of Experimental Medicine (DIMES), University of Genoa, 16126 Genoa, Italy; 3Nephrology and Dialysis Unit, OspedaleBelcolle, 01100 Viterbo, Italy; 4Institute of Pharmaceutical Sciences of Western Switzerland, School of Pharmaceutical Sciences, University of Geneva, 1205 Geneva, Switzerland

**Keywords:** glomerulonephritis, oxidative systems, superoxide dismutase, Annexin family, macrophages

## Abstract

Glomerulonephritis are renal disorders resulting from different pathogenic mechanisms (i.e., autoimmunity, complement, inflammatory activation, etc.). Clarifying details of the pathogenic cascade is basic to limit the progression from starting inflammation to degenerative stages. The balance between tissue injury, activation of protective systems and renal tissue repair determines the final outcome. Induction of an oxidative stress is part of glomerular inflammation and activation of protective antioxidant systems has a crucial role in reducing tissue effects. The generation of highly reactive oxygen species can be evaluated in vivo by tracing the inner-layer content of phosphatidyl ethanolamine and phosphatidyl serine in cell membranes. Albumin is the major antioxidant in serum and the level of oxidized albumin is another indirect sign of oxidative stress. Studies performed in Gn, specifically in FSGS, showed a high degree of oxidation in most contexts. High levels of circulating anti-SOD2 antibodies, limiting the detoxyfing activity of SOD2, have been detected in autoimmune Gn(lupus nephritis and membranous nephropathy) in association with persistence of proteinuria and worsening of renal function. In renal transplant, high levels of circulating anti-Glutathione S-transferase antibodies have been correlated with chronic antibody rejection and progressive loss of renal function. Annexins, mainly ANXA1 and ANXA2, play a general anti-inflammatory effect by inhibiting neutrophil functions. Cytosolic ANXA1 is decreased in apoptotic neutrophils of patients with glomerular polyangitis in association with delayed apoptosis that is considered the mechanism for polyangitis. High circulating levels of anti-ANXA1 and anti-ANXA2 antibodies characterize lupus nephritis implying a reduced anti-inflammatory effect. High circulating levels of antibodies targeting Macrophages (anti-FMNL1) have been detected in Gn in association with proteinuria. They potentially modify the intra-glomerular presence of protective macrophages (M2a, M2c) thus acting on the composition of renal infiltrate and on tissue repair.

## 1. Introduction

Glomerulonephritis (Gn) constitute a heterogeneous group of renal disorders characterized by glomerular inflammation resulting from different pathogenic noxae such as autoimmunity, activation of complement cascade and other factors linked with innate and adaptive immunity. Infiltration by immune cells (neutrophils, T and B cells, etc.) is commonly detected in affected areas of the kidney; the proliferation of intrinsic renal cells (mesangial, Bowman epithelia) occurs in selected conditions. The tubulo-interstitial compartment is also interested in most severe cases, often representing a negative predictor of outcome in terms of progression to renal failure. Different causative triggers usually result in the homogeneity of inflammatory responses and, as direct consequence, the classification of Gn integrate causative elements with pathological findings. Among others, membranous nephropathy-MN, Lupus nephritis-LN and ANCA-associated Gn represent valid examples of renal involvement in autoimmune disease. On the other hand, post-infective, membrane-proliferative and IgA Gn are characterized by renal deposition of immune complexes, not related to autoimmune disease.

In this respect, Idiopathic Nephrotic Syndrome represents a separate entity, where glomerular inflammation resulting in focal areas of segmental sclerosis (FSGS) characterized the late stages of the disease. Common pathological elements justify the administration of similar treatments based on steroids to reduce inflammation and immune-suppressive agents to modulate native and adaptive immune response.

Clarifying details of the pathogenic cascade leading to glomerular and tubulo-interstitial inflammation is fundamental to speculate new therapeutic scenarios that can improve the efficacy of current therapies and limit the progression to advanced stages characterized by glomerulosclerosis and tubulo-interstitial fibrosis. Blocking autoantibody generation through immunomodulatory drugs represent the current therapeutic strategy; targeting specific pathways or key molecules of the inflammatory cascade is may represent a complementary approach.

Activation of the oxidative system follows the deposition of autoantibodies and represents a pro-inflammatory and additive mechanism of renal injury, that should be considered as second potential therapeutic target

In many Gn, antioxidative systems are activated in response to oxidants stress that frequently occur. Components of the Annexins family, are also of interest because they play a direct anti-inflammatory role or mediate the effects of drugs such as corticosteroids [1]. Annexin A1 (ANXA1) and Annexin A2 (ANXA2) are the two molecules most investigated in this area [1].Finally, macrophages (Mφs) play a key regulatory role in the kidney during inflammation, modifying infiltrating cell composition and probably enhancing repair.

At any stage and in all pathological conditions we should consider elements that limit the effects of protective systems or molecules (i.e., antioxidants, ANXA1, ANXA2, anti-Mφs substances) as key molecules participating in disease evolution and should consider the balance between tissue injury, activation of protective systems and renal tissue repair as a determinant of the final outcome [2]. Recent evidence suggests that the efficacy of protective mechanisms and, in parallel, of the renal repair may be reduced in Gn [1,3,4,5] and potentially worsen inflammation.

The description of mechanisms involved in renal tissue protection and repair in Gn and the analysis of factors counterbalancing their effects is the topics of the present review that will consider, in a progressive order, the following points: (a) the equilibrium between oxidative stress and antioxidant systems [6,7], (b) quantitative and qualitative modifications of anti-inflammatory proteins of the Annexin family1 and, finally, (c) Mφs infiltration and effects of anti- Mφs antibodies on the renal outcome. 

The reason for describing such different aspects of Gn is that they contribute synergistically to counterbalance tissue injury pathways and limit their negative effects on renal function [3,4,5].

## 2. Oxidative Systems

Oxidative stress is part of glomerular inflammation in Gn and activation of protective antioxidant systems has a crucial role in counteracting renal injury. Several techniques have been utilized to evaluate both the oxidative and the protective systems.

Cell membrane phospholipids are the major target of reactive oxygen species (ROS) deriving from mithocondria. *Ex vivo* studies on erythrocyte membrane composition in phosphatidyl ethanolamine and phosphatidyl serine content and circulating malondialdehyde levels, a stable product of lipid peroxidation, have been considered as indirect markers of oxidative stress in the past [8,9].

Musante et al. proposed another approach to evaluate the oxidative milieu in serum samples [7]: it is based on the largely recognized antioxidant effect of circulating albumin that interacts with oxidants by means of the unique free sulfhydryl groups (SH)(i.e.,Cys 34) [10]. This free SH is oxidized into a sulfonic group (SO_3−_) and, as consequence, albumin acquires an higher molecular weight (+48 Da) for the adding of three molecules of Oxygen (Figure 1); oxidized albumin has also a more anionic charge resulting from the addition of three negative residues. Due to the high concentration of albumin in serum (50 gr/L), its antioxidant activity is preponderant over other systems. On the basis of the changes above, oxidized albumin can be determined by chemical assays (i.e., titration of SH, evaluation of either the charge and the molecular weight) in serum of patients and probably represents the most reliable marker of oxidative stress ‘in vivo’ [10].

Other antioxidant molecules are involved during systemic inflammation and the reduction of circulating levels for consumption may be correlated with the grade of oxidative stress. Glutathione (GSH) is the key intracellular antioxidant molecule that is almost completely reduced in normal condition: the ratio of oxidized/reduced GSH is considered to be a reliable indicator of intracellular oxidation [8,9]. Glutathione S-transferases (GSTs)are a superfamily of detoxifying enzymes that protect cells from any oxidative stress deriving from inflammation and from the environment by catalyzing the reaction of reduced GSH with oxidants [11]. Catalase and Superoxide Dismutase2(SOD2) are mitochondrial antioxidant enzymes involved in intracellular detoxification: SOD2 transforms superoxide O° into hydrogen peroxide (H_2_O_2_) and diatomic oxygen, Catalase decomposes H_2_O_2_ into water and oxygen and completes the detoxifying cycle [12]. Upon activation of mitochondrial oxidative pathways, SOD is released from mitochondria and externalized by cytoskeletal remodelling [13,14].

### 2.1. Oxidants and Antioxidants in Glomerulonephritis

The status of both oxidant and antioxidant systems has been analyzed in Gn and in FSGS. Overall, studies confirmed that oxidation is part of the inflammatory cascade and also demonstrated that antioxidant systems are rapidly saturated, thus suggesting that loss of their protective effect may contribute to the gravity of the process.

Studies in humans and in animal models have consistently shown that the phosphatidyl ethanolamine and phosphatidyl serine content of erythrocytes is decreased in Gn whereas malondialdheyde, an end product of peroxidation, is high, especially in patients with lupus nephritis, IgA Gn, focal segmental glomerulosclerosis (FSGS) and membranous nephropaty(MN) [6,8,9]; both findings are an indirect proof that oxidative systems are active in Gn. Unfortunately, therapies based on antioxidants utilized in the past had limited effects, probably because the amount of oxidants overload the antioxidative effects of available drugs. This negative outcome has fell short of the expectations on a direct and rapid resolution of the problem in clinical practice.

Important observations that confirmed the high impact of oxidation and the difficulty of limiting its effects in vivo were carried out in FSGS utilizing serum oxidized albumin as reliable marker of oxidative stress activation. Results were of particular interest since they showed that circulating albumin is almost completely oxidized in this condition, thus indicating its high oxidative activity [7]. They also offered the opportunity to make some further considerations relative to the concept that saturation of the antioxidant systems (albumin in this case) is due to a large excess of oxidants that are produced by the system. In fact, it is common clinical experience that any attempt to increase the pool of serum non-oxidized albumin by repeating infusions of exogenous normal albumin do not achieve any positive effects on proteinuria that is a reliable marker of glomerular alterations in FSGS. Albumin is usually re-infused at a rate of 20–30 gr/day that, in a patient weighing 50 Kg with active FSGS and 5 gr day albuminuria, corresponds to 5–6% of the circulating pool of protein; no effect on proteinuria suggests that simple replacement of albumin is not sufficient to temporarily modify the equilibrium between the oxidized and non-oxidized forms. One possible conclusion is that also in FSGS, oxidant generation overloads the amount of albumin replacement (5–6% of the total pool) and justifies the lack of optimism on possible antioxidant therapies utilizing available technologies.

Other indirect observations supporting activation of oxidative pathways in Gn come from studies on detoxifying enzymes. In renal biopsies of patients with MN, Catalase co-localizes with IgG4 along the glomerular basement membrane, clearly showing an extra-cellular localization that is an indirect sign supporting its activation [13]. In MN animal models induced by injecting anti-THSD7A(antigens on podocytes surface), Tomas et al. reported de-localization of SOD2 on podocytes surface at 70 days after disease induction [14]. In the same way, when podocytes were stressed with H2O2 in vitro, SOD2 resulted up-regulated and, after externalization, clustered along the podocyte membrane [14]. In line with the above studies, we recently reported a significant deposition of SOD2 on podocyte membrane of subjects affected by MN3. All these findings support local activation of the detoxifyng system in response to an increased production of oxidants. We have no indication whether the balance between the oxidative and detoxifying systems may limit the negative effects in glomerular cells; however, new hints deriving from studies on anti-SOD2 antibodies in autoimmune glomerular pathologies and in anti-GST antibodies in post-transplant antibody-mediated rejection (see below) seem to allow to us conclude about the extremely negative role of oxidants on renal functional parameters that are indicative of poor outcome.

### 2.2. Anti-SOD2 Antibodies

The protective effect of SOD2 may be reduced by anti-SOD2 antibodies. High levels of circulating anti-SOD2 antibodies have been reported in either MN and Lupus nephritis, two common forms of autoimmune Gn [3,15,16]. In human MN, high levels of circulating anti-SOD2 antibodies represent an independent risk factor of not-remission of proteinuria (<3.5 g/day) and of reduction of renal function (CKD-EPI < 60 mL min) at 1 year of follow up. The risk is maximized when combining the high serum levels of anti-SOD2 and anti-PLA2R1 antibodies: such a finding confirms that the oxidative stress is emphasized by the exposure of glomeruli to the primary antibodies anti-PLA2R1 IgG4 [17]. Similar findings have been reported in subjects affected by Lupus nephritis, where high levels of circulating anti-SOD2 IgG2 were found during the flare of the disease and, according to proteinuria then decreased after effective treatment [16]. Therefore, we speculate that SOD2 may play a role in limiting the oxidative response caused by antibodies against glomerular basement membrane (anti-PLA2R or THSDA7A in MN, anti-dsDNA or anti-enolase in LN).

### 2.3. Anti GST Antibodies

Development of anti-GST antibodies is a part of antibody-mediated rejection (AMR) in recipients of renal graft and is correlated with a loss of renal function, suggesting a role in inducing graft damage [18,19]. It has been hypothesized that anti-GST antibodies are generated by ischemia-reperfusion at the time of the engrafting or by other tissue damaging noxae, such as calcineurin inhibitor toxicity and/or inflammatory processes. It is relevant that the association of anti-GST levels and graft function deterioration was found in absence of modifications of any other antibodies such as HLA-DSAs and in some cases preceded their appearance, implying that HLA-DSAs and anti-GSTT1 are independent predictors of AMR.

## 3. Anti-Inflammatory Mechanisms: The Annexins Family

Annexins are Ca^+^-regulated phospholipid-binding proteins that play an important role in the cell life cycle, exocytosis and apoptosis [20]. Two components of the Annexin family, Annexin A1 (ANXA1) and Annexin A2 (ANXA2), are involved in the glomerular inflammatory response.

ANXA1 is a 37KDa protein expressed in cytoplasm of innate immune cells, such as neutrophils, monocytes, mast cells, macrophages, eosinophils and in minimal amounts in T-cells; ANXA1 is also expressed by some renal components, mainly by Bowman’s capsule, cells of macula densa and medullary/papillary collecting ducts [21,22]. ANXA1 is an anti-inflammatory protein, representing a key modulator of innate immunity [23,24,25]. In more detail, ANXA1 may counteract inflammatory activity at different levels: one—inhibiting the recruitment of neutrophils in inflammatory areas and reducing the adhesion of these cells to endothelia [23,26,27,28]; two—promoting neutrophil apoptosis [26]; and three—acting as a chemoactant for monocytes and stimulating macrophages phagocytosis [29] that remove necrotic debris [26] (Figure 2).

Of interest, ANXA1 expression is induced by glucocorticoids through the Annexin A1 Lipoxin A receptor (ALXR) and the glucocorticod-induced leucine zipper gene, suggesting that their anti-inflammatory effects may be mediated by this protein [30,31]. In line with this, the in vivo model of knock-out mice for ANXA1 is characterized by more severe inflammatory lesions and delayed resolution of inflammatory arthritis and colitis [32,33].

Previous findings emphasized the anti-inflammatory role of ANXA1 in either renal and extra-renal pathologies in humans. The role of ANXA1 in renal conditions, such as acute kidney injury, metabolic renal diseases and renal fibrosis is reported in the literature (for a comprehensive overview see [34]). We will focus below on the involvement of ANXA1 in autoimmune GN, such as ANCA-associated polyangitis [35] and LN [4], that offers us the opportunity to consider different pathogenetic implications of the protein.

ANXA2, the second molecule of interest, is expressed by several cells and has distinct functions. Its main role is the maintenance of vascular integrity that regulates chemotaxis of neutrophils. By binding the p11 factor (also known as S100A10), ANXA2 interacts with circulating plasminogen and tPA and activates plasminis, thus having an important fibrinolytic effect [36]. As such, it contributes to regulate inflammation and the immune system [37] by achieving digestion of extracelluar matrix in membranes, limiting chemotaxis of macrophages [38,39]. Mice lacking ANXA2 (*Anxa2*^−/−^) develop two important phenotypes linked with the functions of the protein: in respect to inflammation, *Anxa2*^-/-^ manifests a more severe phenotype [40]; in respect to the fibrinolytic function, *Anxa2*^-/-^ mice accumulate fibrin in several organs, including the kidney, and have an impaired removal of vascular thrombi similar to what observed in plasminogen, uPA and tPA deficient mice [41].

There are at least three different modalities according to which the protective role of both ANXA1 and ANXA2 may be modified in Gn. The first is quantitative reduction of protein expression in relevant cells (i.e., neutrophils in polyangiatis), the second is correlated with post-translational modifications such as deimination occurring in ANXA1 of neutrophil extracellular traps (NETs) in SLE and the third is the formation of autoantibodies that target both ANXA1 and ANXA2.

### 3.1. Neutrophyl ANXA1 in Granulomatosis with Polyangiitis in Kidney

In kidney, granulomatosis with polyangiitis [35] are characterized by the proliferation of cells of the renal micro-vascular system, often associated with serious glomerular inflammation; circulating-protease 3 antibodies (anti-PR3) are the hallmark of the disease. In granulomatosis with polyangiitis, neutrophils are either the target of autoimmunity (anti-PR3 antibodies) and are also potential effectors of the glomerular microvascular damage, probably due to a delayed apoptosis that extends the presence of these cells in glomeruli. Everts-Graber et al. [35] recently showed that cytosolic ANXA1 is markedly decreased in the apoptotic neutrophils of patients with glomerular polyangitis and is almost completely delocalized from the cytosol to the membrane. In apoptotic neutrophils, the externalization of ANXA1 correlates with the externalization of phospholipid scramblase 1 (PLSCR1), a substance that increase the cell life and seems implicated in the autoimmune process. Reduced intracellular levels of ANXA1 reinforce the effect of PLSCR1 in delaying apoptosis, that is considered one of the pathogenic mechanisms in granulomatosis with polyangiitis. The significance of delocalized ANXA1 on the membranes of neutrophils is not clear and requires further investigation.

### 3.2. Annexin A1 in Rheumatoid Arthritis (RA) and in SLE

Studies in RA documented high levels of Annexin A1 in fibroblasts recovered from synovial fluids [42] where this protein plays an anti-inflammatory effect correlated with the cytokine milieu and with response to steroids. Annexin A1 is also overexpressed by circulating neutrophils in SLE by a factor of 2–3 compared to control cells and is a component of NETs whose formation in patients with SLE has been associated with the activation of autoimmunity since within NETs, proteins and DNA undergo post-translational modifications [43]. In 67% of patients with Lupus nephritis, also Annexin A1 was shown to be modified for the presence of 1 Citrulline in place of Arginine 188 (peptide N178-L198) (Figure 3). In the unique report on circulating levels, Annexin A1 has been reported to be high in a large cohort of 219 patients with SLE, half of which had Lupus nephritis. Free ANXA1 did not correlate with any clinical variables in SLE patients. The above observations suggest that Annexin A1 is more dynamically and structurally modified in response to the inflammatory milieu in tissues and cells than in circulation, representing a possible mechanism for the generation of auto-antibodies.

### 3.3. Anti-ANXA1 Antibodies in Lupus Nephritis (LN)

Anti-lipocortin 1 antibodies, homologous of ANXA1, have previously been proposed as a marker [42] of Systemic Lupus Erythematosus (SLE) activity [44]. In 2010, the presence of circulating anti-ANXA1 antibodies has been reported in association with discoid lupus [45] and, more recently, in patients with LN [46,47]. Anti-ANXA1 IgG2 have been also micro-eluted from glomeruli of patients with LN, suggesting a direct implication in the renal process [4]. Overall, anti-ANXA1 antibodies may act with a double effect: they reduce the amount of protein in circulation and limit its anti-inflammatory role (first mechanism) and also deposit in glomeruli acting as a direct autoantibodies for the kidney(second mechanism). A third mechanism is only hypothetical and it is based on the finding that serum level of circulating anti-ANXA1 IgG2 directly correlated with circulating anti-dsDNA IgG2 that is a class of antibodies implicated in LN. We hypothesized a mimicker effect of dsDNA for ANXA1: both molecules (ANXA1 and dsDNA) share, in fact, an anionic epitope and the same pH-dependent binding kinetics (minimal at pH 4 and maximal at pH9–10). Competition experiments indicated that at the physiologic pH of blood (7.4), dsDNA has stronger affinity for anti-ANXA1 antibodies than ANXA1 and competes with ANXA1 for the binding with Anti-ANXA1 IgG2 on a charge-mediated base [48]. This means that an amount of anti-dsDNA IgG2 actually exists in anti-ANXA1 antibodies, reinforcing the suggestion of a direct toxic effect in glomeruli.

### 3.4. ANXA2 Antibodies in SLE, Antiphospholipid Syndrome and Nephrotic Syndrome

As previously described, ANXA2 is a profibrinolytic endothelial cell surface receptor that binds beta(2)-glycoprotein I (beta2GPI), the main antigen for antiphospholipid antibodies. Caserman-Maus et al. [49] described that anti-ANXA2 antibodies contribute to the prothrombotic effects in antiphospholipid syndrome (APS). In line with this effect, anti-ANXA2 IgG were also associated with the increased risk of thrombosis and/or pregnancy morbidity patients with APS and SLE [50,51].

More recently, Ye et al. [52] described anti-ANXA2 IgG in serum and in renal tissue of patients affected by NS. Therefore, authors speculated that patients with histological findings of minimal change lesions and deposition of IgG along the glomerular basement membrane suggests an autoimmune condition.

## 4. Macrophages, Tissue Infiltration and Repair

Macrophages (Mφs) are mononuclear cells characterized by sub-phenotypic heterogeneity that reflects a variety of functions. Mφs (CD68+/CD11b+) originates in the bone marrow [53] and migrates differentiating in dendritic cells (DCs), Mφs Type I (M1) or Type II (M2). Such Mφs-derived cells present different surface markers and are devoted to different functions. DCs (CD11c+) are deputed to immuno-surveillance mostly as antigen-presenting cells. During inflammation, Mφs further differentiate into four different cell sub-types with different functions: (a) M1(CD38+) are stimulated by Pathogen-associated Molecular Patterns (PAMP) and Damage-associated Molecular Patterns (DAMP) and are involved in regulating inflammation and repairing mechanisms [54], (b) M2a (CD206+) are stimulated by IL-4/IL-13 and are involved in wound healing, (c) M2b (CD86+) are stimulated by IL-1b and LPS and are involved in immune regulation and, (d) M2c (CD163+), stimulated by IL-10 and TGFB, regulate tissue repair. Therefore, Mφs-derived cells act with different functions, varying from pro-inflammatory (M1) to immune-regulation (M2b) and tissue repair (M2c).

### 4.1. Macrophage Implication in Gn

Evidence for the involvement of macrophages in Gn is scarce and mostly based on the characterization renal infiltrates in human and in animal models of Gn. The grade of infiltration is not directly related to the grade of activity, therefore our knowledge is limited to a mere description of the presence/absence of the different monocytic cell phenotypes detected in the Gn. Moreover, only in the last two decades was CD68 recognized as the main marker of macrophages and relied essentially on anti-FM32 and anti-Leu 32 for monocyte characterization [55,56,57] (for a more comprehensive revision of the literature, please refer to [58]). With the limitation above, studies demonstrated a correlation between macrophage infiltration and severity of renal damage, both in rodents and humans with IgA Gn and Lupus nephritis [55,59,60]. However, new studies describing the renal infiltrates based on new cell markers are needed, to compare pro-inflammatory macrophages subtypes with others involved in repair.

The in vivo model of knock-out mice for macrophages showed a lower grade of tissue lesions and better renal function compared to controls [61,62,63,64]. However, a more specific experimental deletion of M1 or M2 is not available. Based on functions of the different Mφs subtypes and on the improvement of tissue lesions and kidney function in that model, the absence of pro-inflammatory M1 cells may be speculated.

In line with this, in murine models of glomerulosclerosis induced by adriamycin and in streptozocin diabetic nephropathy, infusion of M2a and M2c reduced cytokines recruitment and ameliorates both the structural and functional renal injury [65,66]. More recently, in vivo studies of renal damage showed a prominent effect of M2c over M2a in the reduction of infiltration and fibrosis, proposing a regulatory effect of M2c over other immune cells. In more detail, M2c are able to stimulate and induce stimulation of Treg in both in vivo and in vitro experiments, suggesting that the M2c effect is mediated by Treg [61,65,67].

Therefore, given the contrasting of effects of the different macrophage subsets (M1 vs. M2), Mφs are to be considered as fundamental regulator of inflammatory response in the kidney.

### 4.2. Anti-Macrophage Antibodies in Gn

We recently reported circulating anti-Mφs antibodies in serum of patients with autoimmune MN [5,68]. Such antibodies recognized a protein of the formin-like family, FMNL-1, expressed by podosomes of Mφs. FMNL-1 was found to be fundamental in regulating the Mφs diapedesis and in regulating the synapses between Mφs and immune cells [69,70]. Through a home-made ELISA for FMNL-1, we demonstrated that a large amount of subjects with MN had very high levels of circulating anti-FMNL1 IgG4. Moreover, the highest titre correlated with worst clinical outcome. More studies are needed to further investigate the role and the genesis of anti-FMNL1 antibodies.

We need to better explore the correlation between serum levels, kidney inflammation in terms of quantity and quality, and renal function. Preliminary data in patients with LN are confirming the direct correlation between the high serum levels of anti-FMNL1 antibodies and proteinuria. In vivo experiments blocking the effect of anti-FMNL1 antibodies would be useful to better understand such correlations.

## 5. Concluding Remarks

While it is widely accepted that autoimmunity, inflammation, oxidants and other molecules are involved in the pathogenesis of Gn, it is less considered that other mechanisms limit the deleterious effects of all pathogenetic triggers active in Gn. The balance between the proactive and the limiting factors determine the final outcome of renal injuries in many Gn. Loss of albumin (the main antioxidant molecule) in urine and almost complete oxidation of serum albumin in patients with nephrotic syndrome are probably deleterious for the kidney. The presence of antibodies against major antioxidant enzymes or against proteins involved in the immune response such as ANXA1 and ANXA2 have been reported in different Gn. Antibodies targeting macrophages in patients with MN have been described as well, implying also a potential limitation of the role of these cells in the inflamed kidney. It is time to consider that factors limiting the immune response or the tissue repair that follows inflammation may be considered in the pathogenesis of auto-immune and inflammatory conditions with the same importance as other causative mechanisms.

## Figures and Tables

**Figure 1 ijms-24-08318-f001:**
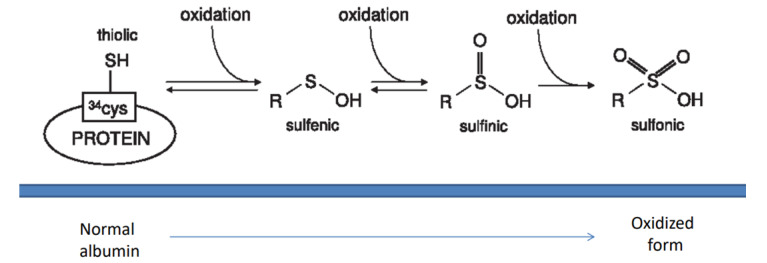
Scheme for oxidation of serum albumin. Oxidants react with the unique free SH in position 34 of the protein structure that add in sequence 3 Oxygen residues that increase the MW of the protein (+48 daltons) and modify the charge of albumin (for a review see [10]). The final molecule contains a sulfonic acid in place of the original.

**Figure 2 ijms-24-08318-f002:**
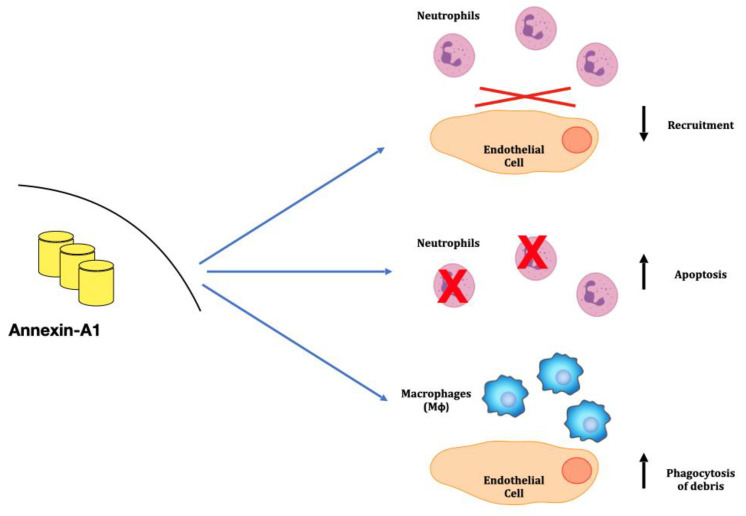
Schematic representation of Annexing A1 effects. Activation of intracellular Annexin A1 in immune cells promotes anti-inflammatory effects.

**Figure 3 ijms-24-08318-f003:**
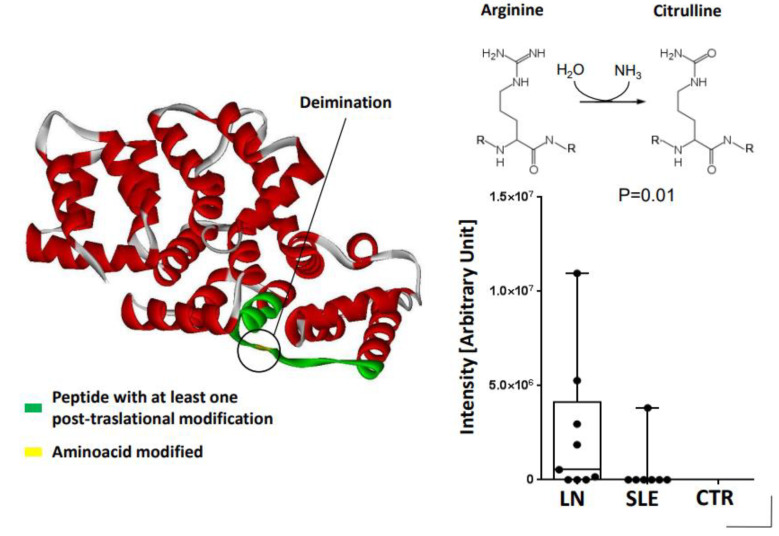
Chemical modifications of Annexin A1 purified from NETs of patients with Lupus nephritis. The modified protein is an Annexin A1 modified for the presence of 1 Citrulline in place of Arginine 188 (peptide N178-L198). The amount of the modified peptide (N178-L198) is increased in Annexin A1 purified from NETs of patients with Lupus nephritis compared to SLE and normal subjects.

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
