# Peer review of "Mechanisms Limiting Renal Tissue Protection and Repair in Glomerulonephritis"

_ijms, 2023, doi:10.3390/ijms24098318_

Round 1

Reviewer 1 Report

This manuscript summarizes the role of oxidative stress, apoptosis, and macrophage (Mφ) in the balance between tissue damage and repair in glomerulonephritis.

The manuscript is well described in a reasonable manner, but I think it would be more attractive if an easy-to-understand figure was inserted for readers. Please provide each figure about the role of oxidative stress, the role of annexins, and the role of Mφ in glomerulonephritis. In your previous paper (Ref. 58), you provided several easy-to-understand diagrams of Mφ.

Author Response

We many thanks the Reviewer for the positive comments.

We here provide some figures, as well suggested

Reviewer 2 Report

The authors have done an excellent job of analyzing the mechanisms of renal inflammation and raising questions about the "holistic approach" applied so far. The article describes the molecular details and oxidative mechanisms well, consistent with the nature of the journal.

 I would suggest some modifications to increase its quality even more.

1. Page 2. “Different causative triggers usually result in homogeneity of inflammatory responses and, as direct consequence, the classification of Gn integrate causative elements with pathological findings: autoimmune (Membranous-MN, Lupus nephritis-LN), immune complex Gn (post-infectious, membrane proliferative) and IgA Gn are the major groups, followed by C3, pauci-immune and ANCA associated Gn." This pathogenetic classification is quite unusual: ANCA Gn is out of the autoimmune group just as IgA Gn appears to be out of the Gn group of immune complexes. Is this what the Authors meant? Are there any references on this grouping?

2. Section 2.3 “Antibody-mediated reaction”. Do the authors mean "rejection"?

3. Page 6. "...may be modified in immunologic and autommine glomerulonephritis..." The authors use different terms and groups than those described on page 2. This could lead to confusion.

4. Paragraph 3.1. Microscopic polyangiitis is usually driven by MPO while granulomatosis with polyangiitis is predominantly PR3 as reported by the cited reference. The term "ANCA polyangiitis" used is a bit too generalized and quite unusual.

5. Paragraph 4: Typo in the title ("an");  DCs stands for dendritic cells? Overall, this paragraph needs to be more schematic, the sentences less long.

6. Paragraph 4.1. Two sentences with a different character are extremely similar/equal to those already published by some same Authors in the reference given. I'd ask for them to be reworked, even though the authorship is largely, but not all, the same.

7. Some sentences in different paragraphs seem to have different characters. Please check.

Author Response

a
